# Application of Life-Dependent Material Parameters to Fatigue Life Prediction under Multiaxial and Non-Zero Mean Loading

**DOI:** 10.3390/ma13071587

**Published:** 2020-03-30

**Authors:** Krzysztof Kluger, Aleksander Karolczuk, Szymon Derda

**Affiliations:** Department of Mechanics and Machine Design, Opole University of Technology, ul. Mikołajczyka 5, 45-271 Opole, Poland; a.karolczuk@po.edu.pl (A.K.); szymon.derda@doktorant.po.edu.pl (S.D.)

**Keywords:** multiaxial fatigue, fatigue life prediction, mean value of stress, life-dependent material parameters, S355 steel, 7075-T651 aluminium alloy

## Abstract

This study presents the life-dependent material parameters concept as applied to several well-known fatigue models for the purpose of life prediction under multiaxial and non-zero mean loading. The necessity of replacing the fixed material parameters with life-dependent parameters is demonstrated. The aim of the research here is verification of the life-dependent material parameters concept when applied to multiaxial fatigue loading with non-zero mean stress. The verification is performed with new experimental fatigue test results on a 7075-T651 aluminium alloy and S355 steel subjected to multiaxial cyclic bending and torsion loading under stress ratios equal to *R* = −0.5 and 0.0, respectively. The received results exhibit the significant effect of the non-zero mean value of shear stress on the fatigue life of S355 steel. The prediction of fatigue life was improved when using the life-dependent material parameters compared to the fixed material parameters.

## 1. Introduction

Non-monotonic alternation of stress states may lead to permanent changes in the structure of materials and is often the cause of the limited functionality of machines and engineering structures [1,2,3,4]. The mechanisms leading to fatigue failure of materials are complex and depend on many factors [5]. To avoid multiscale damage modelling of material behaviours, phenomenological models are mostly applied for fatigue life prediction in engineering problems [6]. These models are usually based on a semi-empirical function of multiaxial stress/strain components correlated with fatigue life [7,8,9,10]. Unfortunately, fatigue life prediction models are very often limited; i.e., limited to a given type of material, stress path, temperature, fatigue life range, etc. There is a tendency to modify such models in order to extend their scope of operation. This has resulted in the development or modification of many models over the last few decades [10,11,12,13,14,15,16]. Functions that reduce the fluctuation of multiaxial stress states to an equivalent scalar value are an integral part of fatigue models. In the fatigue life calculation algorithm, this scalar value is compared with the corresponding reference characteristics, resulting in fatigue life estimation.

Among many criteria described in the literature, a group can be distinguished which is characterised by the assumption that components of the stress/strain vector in a material plane with a specific orientation are correlated with the fatigue life. This proposal, called the critical plane approach, has gained great interest [17,18,19,20,21,22,23,24,25,26,27,28]. Depending on the material, strain state, environment, component geometry, and stress amplitude, the fatigue process may be dominated by cracking in the plane of maximum shear or normal stresses [29,30]. For this reason, many researchers formulate fatigue criteria that are dependent on the crack mode, and the criterion itself is a certain combination of normal and shear components of the stress/strain vector in the critical plane. The stress-based criteria usually feature linear or non-linear functions of material parameters and shear *τ_ns_*, normal *σ_n_* (in the critical plane), or hydrostatic *σ_h_* (stress invariant) stresses. The criteria proposed in the literature most often assume that material parameters used in the multiaxial stress-reducing function are constant, and serve to balance the different effects of normal and shear stress components on the development of damage in the material. Their values can be estimated on the basis of experimental data, mainly for uniaxial loading. Fatigue criteria, in their original form, are usually used to assess the limit state called the fatigue limit [31,32,33,34]. Therefore, the material parameters are a function of the fatigue limits from uniaxial fatigue tests. The studies carried out by Socie [29] have shown that the dominant mechanisms of damage depend not only on the loading and type of material but also on the fatigue life. Socie [35,36] noted that for stainless steel 304 and Inconel 718 and 1045, the crack nucleation period is controlled by shear stress, which increases with fatigue life when compared to that controlled by normal stress. Based on these observations, it can be concluded that the material parameters assessing the effect of shear and normal stress on fatigue life are not constant, but instead dependent on life. The phenomenon of the life dependency of the material parameters in multiaxial fatigue models has been analysed in [21,37,38,39,40,41]. It was demonstrated that the life-dependent material parameters can be successfully applied to multiaxial stress- and strain-based fatigue life prediction models under cyclic proportional and non-proportional loading paths with zero mean stress. The variability of the material parameters over the fatigue life range strongly depends on the uniaxial fatigue stress/strain–life curves. For materials with parallel uniaxial fatigue curves, the material parameters are independent of the fatigue life, and as a result, they can be considered as fixed.

Additional static loading could be an important factor affecting fatigue life [42,43]. A non-zero mean value of the stress is often caused by the effect of self-weight of the working element as well as the effect of initial tension in load-bearing elements (e.g., belts in gearboxes) or residual stresses [44,45,46,47]. The effectiveness of multiaxial fatigue models with life-dependent material parameters under cyclic multiaxial with zero mean stress has been validated in a few research papers [21,37,39]. For fatigue loading with a non-zero mean stress, the life-dependent material parameters concept was analysed only in [48], which focused on the algorithm of lifetime calculation, including the concept of life-dependent material parameters. The proposed concept needs to be further verified with a larger group of materials and models.

The aim of this work is to verify the life-dependent material parameters concept when applied to the fatigue life prediction of a 7075-T651 aluminium alloy and S355 steel under multiaxial fatigue loading with a non-zero mean stress. The results of the new experimental fatigue tests under uniaxial and multiaxial cyclic bending and torsion loading with stress ratios equal to *R* = −0.5 and 0.0, respectively, are presented here. Several well-known stress-based fatigue life prediction models are implemented. For the first time, the life-dependent material parameter concept has been introduced to the Crossland [49], Papadopoulos [50,51], and Papuga–Růžička [52] criteria under fatigue loading while including a non-zero mean stress.

## 2. Multiaxial Fatigue Life Models with Life-Dependent Material Parameters

The life-dependent material parameters concept analysed in this paper concerns the wide group of multiaxial fatigue life prediction models with the following shared assumptions; namely, (i) there is a scalar function F of spatial stress/strain components $\sigma_{ij}\left(t\right),\varepsilon_{ij}\left(t\right),\;$ and the material parameters, K, are equivalent to the stress/strain state under uniaxial loading (equivalent with respect to the fatigue life); (ii) the fluctuating stress/strain histories $\sigma_{ij}\left(t\right),\;\varepsilon_{ij}\left(t\right)$ could be replaced by the amplitudes and mean values of counted loading cycles. Most of the fatigue models based on the critical plane approach including stress-based, strain-based, and energy-based models [7,53], but also models based on the stress invariants [54] belonging to this group. The general equation for fatigue life prediction can be presented as follows:(1)$$F\left(\sigma_{ij}\left(t\right),\varepsilon_{ij}\left(t\right),K\right)=q\left(N_{f}\right)$$
where $q\left(N_{f}\right)$ is the reference fatigue characteristic, relating the stress/strain amplitude with the number of cycles to failure $N_{f}$ under uniaxial cyclic loading. The material parameters K are identified by applying Equation (1) to uniaxial loading. Most fatigue models in the analysed group distinguish shear and normal stress/strain components in the critical plane or deviatoric and hydrostatic stress invariant components to be crucial for determining the fatigue failure of metallic materials. As a consequence, the material parameters K serve as the weighting factors of shear and tensile selected stress/strain components. For a wide range of cycles to failure, ***K*** is not a constant value as the mechanisms of fatigue failure depend essentially on the value of the applied load. The variability of ***K*** can be demonstrated by identifying its value using uniaxial loading. For example, for the applied uniaxial stress state with amplitude $\sigma_{f}$ leading to failure at $N_{f}$ number of cycles, Equation (1) with the reference S–N curve for torsion loading $q\left(N_{f}\right)=\tau_{f}\left(N_{f}\right)$ can be shown as:
(2)$$F\left(\sigma_{f}\left(N_{f}\right),K\right)=\tau_{f}\left(N_{f}\right).$$

The parameter ***K*** can be found by solving the above equation relative to ***K***, obtaining the function *f* in its general form as follows:(3)$$K=f(\sigma_{f}(N_{f}),\tau_{f}(N_{f})).$$

The same identifying procedure should be applied to the strain- and energy-based fatigue models that lead to conclusions of the life dependency of material parameters when introduced as weighting factors into the stress/strain reducing function F.

## 3. Stress-Based Multiaxial Fatigue Models

To verify the proposed methodology for fatigue life calculation covering the variability of material parameters depending on the number of cycles to failure, fatigue models were selected with significant differences in stress-reducing function and definition of the critical plane. Since the selected models are well documented in many papers, only a brief summary is given below. The application of material fatigue models for fatigue life estimation consists of calculating the equivalent stress value *σ_eq_* and comparing it with the material S–N curve. In the analysed models, the material parameters are mainly a function of the ratio of torsion fatigue strength to axial fatigue strength; i.e., $r_{\sigma }\left(N_{f}\right)=\tau_{f}\left(N_{f}\right)/\sigma_{f}\left(N_{f}\right)$. The fatigue models and their life-dependent material parameters are shown in Table 1. The first column in Table 1 presents the selected semi-empirical stress-based functions for fatigue life calculation in the form of equations with the life-dependent material parameters (the names of inventors are also included). The second column presents the formulae of the life-dependent material parameters applied to a particular model. Among the seven selected models, there is one based on stress invariants (the Crossland model) and six critical plane approach models. The critical plane orientation is determined as the plane of maximum shear stress (Matake, Dang Van, and Papadopoulos models), the plane with maximum equivalent stress value (Stulen–Cummings–Findley and Papuga–Růžička models), or the plane shifted by the δ angle with respect to the average principal stress directions (Carpinteri–Spagnoli model). The mean value of loading is taken into account in the presented models, expect for the Matake one, by superimposing the mean value to the amplitude of normal or hydrostatic stresses. In the Matake model, the mean value is not included.

In [60], Carpinteri et al. found that static normal stresses imposed on cyclically changing normal stresses significantly reduce fatigue strength, while mean shear stresses do not affect fatigue life. The modification adopted in [60] comprises the implementation of the Goodman [61] model to the Carpinteri–Spagnoli criterion [59] and takes the following form:(4)$$\sqrt{{\left(\sigma_{n,a}+\sigma_{f}(N_{f})(\frac{\sigma_{n,m}}{\sigma_{u}})\right)}^{2}+k_{CS}{\left(N_{f}\right)}^{2}\tau_{ns,a}^{2}}-\sigma_{f}(N_{f})=0.$$

In this paper, it is proposed that Morrow’s correction be included [62], which is applied to the mean value of the normal stress. The result is that the mean stress is reduced by the axial fatigue strength coefficient $\sigma_{f}^{’}$ and the fatigue model can be described with the following equation:
(5)$$\sqrt{{\left(\sigma_{n,a}+\sigma_{f}\left(N_{f}\right)\left(\frac{\sigma_{n,m}}{\sigma_{f}^{’}}\right)\right)}^{2}+k_{CS}{\left(N_{f}\right)}^{2}\tau_{ns,a}^{2}}-\sigma_{f}\left(N_{f}\right)=0.$$

## 4. Experimental Research

Experimental tests for the analysed materials were carried out under the load of constant amplitude plane-bending moment $M_{b}$, torsion moment $M_{t}$, and two combinations of proportional bending and torsion with ratios of applied stress amplitudes of $\tau_{a}=1.0\sigma_{a}$ and $0.5\sigma_{a}$. The plane bending and torsion were generated by a single force applied to lever with a length of 0.2 m. The rotation of specimen by angle β with respect to the lever decomposes the principal moment into bending and torsion moments with a fixed ratio for a given angle β. Details of the applied fatigue stand can be found in [63]. The applied frequency of loading varied between 5 and 30 Hz, depending on the applied stress amplitudes. The tests included loading with the stress ratio *R* equal to −1, −0.5, and 0. A specimen stiffness drop equal to 20% defines the fatigue failure. Figure 1 shows the geometry of the specimen used for testing.

S355 steel specimens were manufactured from a rolled bar with a diameter of 22 mm in the as-delivered condition. The chemical composition and basic strength properties are shown in Table 2. Aluminium alloy 7075-T651 specimens were manufactured from an extruded bar with a diameter of 16 mm in the as-delivered condition. The chemical composition and basic mechanical properties are presented in Table 3 and Table 4, respectively.

The results of the experimental studies were used to determine the uniaxial fatigue characteristics (S–N curves) in the following form:(6)$$\log\left(N_{f}\right)=A-m\cdot \log\left(\sigma \right).$$

For cyclic bending and torsion, where *A* and *m* are linear regression factors, σ is the amplitude or maximum value of the stress in the cycle according to the selected loading (Table 5). Fatigue characteristics for fully reversed bending and torsion (*R* = −1) and for zero-pulsating bending (*R* = 0) along with a fatigue scatter band with $T\left(0.95\right)$ [64,65] are presented in Figure 2a for S355 steel and in Figure 3a for 7075-T651 aluminium alloy. The fatigue scatter band T for each experimental fatigue life $N_{exp}$ was calculated by applying the following formula:(7)$$T=\left\{\begin{array}{ccc} \frac{N_{exp}}{N_{f}} & for & N_{exp}\geq N_{f}\\ \frac{N_{f}}{N_{exp}} & for & N_{exp}<N_{f} \end{array}\right..$$

The empirical cumulative distribution of T was used to calculate $T\left(0.95\right)$ for a 0.95 probability. The shape-preserving piecewise cubic interpolation was applied to find T at 0.95 probability. The value of $T\left(0.95\right)$ presents the required fatigue scatter band around the fitted S–N curve to include 95% of data. In addition, Figure 2b (for steel S355) and 3b (for aluminium alloy 7075-T651) show graphs of the ratio of fatigue strengths in fully reversed torsion to axial stress at *R* = −1 and 0. The range of the $r_{\sigma }$ ratio for the analysed number of cycles was $\Delta r_{\sigma }=0.21$ for steel S355, and $\Delta r_{\sigma }=0.37$ for aluminium alloy 7075-T651. The variation in $r_{\sigma 0}$ is related to the zero-pulsating bending loading, and for both steel and aluminium, the variation was insignificant (red dashed lines in Figure 2b and Figure 3b).

The variability of the normalised life-dependent material parameters applied in the multiaxial fatigue models is shown in Figure 4 and Figure 5. The maximum values of each material parameter along with their ranges, normalised by maximum value, are included in the legends of Figure 4 and Figure 5. The highest relative change in the material parameters was observed for the Crossland model for both tested materials.

Exemplary photos of macroscopic fatigue cracks in S335 steel and 7075-T651 aluminium alloy are presented in Figure 6 and Figure 7, respectively. Dye penetrant inspection was applied to reveal the surface cracks presented in Figure 6 and Figure 7. The macroscopic cracks on the specimen surfaces are perpendicular to the applied stress under cyclic bending and independent of the number of cycles for the failure of S355 steel (Figure 6a). However, under cyclic torsion loading, the macroscopic crack mode depends on the number of cycles for failure. For shorter fatigue lives below 30,000 cycles, a pure shear mode was observed (Figure 6b). Increasing the number of cycles to failure, cracks under the shear mode became shorter, along with additional tensile crack mode. Above 400,000 cycles for failure, only a tensile macroscopic crack mode was observed (Figure 6b). Those observations of the crack mode, depending on the number of cycles for failure, could explain the life dependency of the material parameters analysed in the present paper.

For the 7075-T651 aluminium alloy, the macroscopic cracks under cyclic bending observed on the specimen surface are perpendicular to the applied stress. Multiple cracks were observed for shorter fatigue lives (Figure 7a), while a single dominant crack was observed for lives above 10,000 cycles. Under cyclic torsion, the pure shear crack mode was observed for fatigue lives of below 600,000 cycles to failure (Figure 7b). An additional tensile crack mode was noted in specimens with fatigue life above 600,000 cycles (Figure 6b).

Variation of dominant damage mechanism (crack mode or/and number of initiated cracks) depending on the number of cycles to failure results in the life-dependent ratio of fatigue strengths $r_{\sigma }\left(N_{f}\right)$**,** and as a consequence, the material parameters applied in multiaxial models must also be life-dependent. All experimental results are included in Appendix A.

## 5. Results and Analysis

The performance of the life-dependent material parameters applied to the fatigue life models was assessed on the basis of the fatigue scatter band T (Equation (7), where $N_{f}$ was replaced by $N_{cal}$) at a confidence level of 0.95 [21,64,65]. Its value presents the required uniform scatter band around the perfect agreement $N_{exp}=N_{cal}$ to include 95% of data. A lower value of the coefficient $T\left(0.95\right)$ exhibits improved correlation for the experimental and calculated fatigue lives. The experimental fatigue scatter bands presented in Figure 2a and Figure 3a, which were determined for uniaxial loading, must be used as reference values for the proper validation of multiaxial fatigue models.

The number of cycles to failure, $N_{cal}$, was calculated using two methods; namely, those with fixed and life-dependent material parameters. For the first method, the material constants were calculated from the S–N curves at a reference number of cycles equal to 2 × 10^6^.

Due to the large number of calculation results, only two exemplary figures (Figure 8 and Figure 9) with the comparison of the experimental and calculated fatigue lives are presented. Each legend of Figure 8 and Figure 9 shows the fatigue scatter $T\left(0.95\right)$ calculated separately for each type of loading, and the overall scatter band $T\left(0.95\right)$ is presented in upper-left corner of the plot. The dashed lines indicate a scatter band with a value equal to 3 and the continuous line indicates a perfect match. Figure 8 and Figure 9 show the percentages of the conservative results, where $N_{cal}<N_{exp}$. Those exemplary presented figures present the results obtained with the application of the Crossland model. In general, similar characteristics of the results were obtained for all the applied fatigue models. As a result of zero hydrostatic stress under pure torsion loading, the value of the material parameters $k_{C}$ does not affect the equivalent stress in the Crossland model. Thus, the points in the $N_{exp}-N_{cal}$ plots (Figure 8 and Figure 9) under pure torsion are located in the same position, independent of the applied material parameters concept. According to the results obtained under pure torsion, the non-zero mean value of shear stress has a significant influence on the fatigue life for S355 steel and less of an effect for the 7075-T651 aluminium alloy. This effect was not taken into account in the analysed fatigue models, for which only the non-zero mean value of the normal or hydrostatic stress influences the fatigue life.

A summary of calculation results for all types of loading is presented in Figure 10, Figure 11, Figure 12 and Figure 13. The values of the $T\left(0.95\right)$ coefficient for all analysed models using the fixed and life-dependent values of material parameters are presented in panel (a) of Figure 10, Figure 11, Figure 12 and Figure 13. Panel (b) of Figure 10, Figure 11, Figure 12 and Figure 13 present the change of $T\left(0.95\right)$ (i.e., $T\left(0.95\right)<0$ denotes an improvement of the fatigue life prediction due to the application of the life-dependent material parameters).

Due to the application of the life-dependent material parameters, the calculated fatigue lives better correlate with the experimental fatigue lives for all analysed fatigue models.

For S355 steel, the lowest value of $T\left(0.95\right)$ was obtained for the Papuga–Růžička model; i.e., $T\left(0.95\right)=3.3$ under R=0.0 and $T\left(0.95\right)=4.5$ under R=−0.5. However, those values are still unsatisfactory if compared to the experimental value of the fatigue scatter band equal to 1.9 (Figure 2a). This is mainly due to the apparent effect of the non-zero mean shear stress on the fatigue life for S355 steel, which is not taken into account in the models analysed here.

For the 7075-T651 aluminium alloy, the effect of a non-zero shear stress is pronounced for higher mean values, i.e., pronounced under R=0.0, whereas the effect when R=−0.5 is insignificant. Under the stress ratio R=0.0, the lowest value of $T\left(0.95\right)$, equal to 8.7, was received for the Dang Van and Papadopoulos models with the life-dependent material parameters (Figure 12). The value of 8.7 is unacceptable, since it is more than three times higher than the experimental scatter, equal to 2.4 (Figure 3a). For a lower value of mean stress, i.e., under stress ratio R=−0.5, better consistency of the calculated and experimental fatigue lives was obtained (Figure 13). The lowest value of $T\left(0.95\right)$, equal to 3.1, was received for the Crossland model (see also Figure 9b) with the application of the life-dependent material parameters. The value of 3.1 is only 30% higher than the experimental value, which is equal to 2.4 (Figure 3a).

The inclusion of Goodman’s (Equation (4)) and Morrow’s (Equation (5)) corrections in the Carpinteri–Spagnoli model for S355 steel and 7075-T651 aluminium alloy improved the correlation of results under the stress ratio R=0.0 with respect to the basic form of the model, while under ratio R=−0.5, all three model forms give similar results.

## 6. Conclusions

The following conclusions can be drawn from our analyses:The application of the life-dependent material parameters improved the consistency of the experimental and calculated fatigue lives for all analysed fatigue models and for both S355 steel and 7075-T651 aluminium alloy under non-zero mean stress.The best (though still unsatisfactory) consistency of the experimental and calculated fatigue lives was obtained for the Papuga–Růžička model for the S355 steel for both stress ratios of *R* = 0 and −0.5.The best and satisfactory consistency of the experimental and calculated fatigue lives was obtained for the Crossland model with the life-dependent material parameters for the 7075-T651 aluminium alloy with a stress ratio of *R* = −0.5.The reduction of mean stress with Goodman’s and Morrow’s corrections in the Carpinteri–Spagnoli model resulted in insignificant improvement of the calculation results in comparison with the experimental ones.A non-zero shear stress has a pronounced effect on the fatigue life of S355 steel for both stress ratios but shows an insignificant influence for the 7075-T651 aluminium alloy under a stress ratio of *R* = −0.5.

## Figures and Tables

**Figure 1 materials-13-01587-f001:**
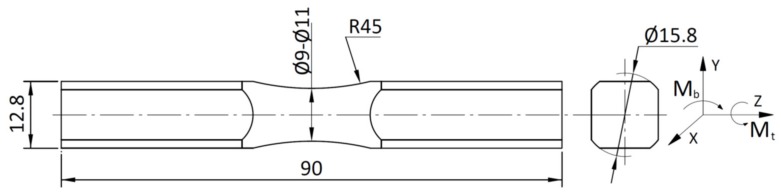
Geometry of the specimen.

**Figure 2 materials-13-01587-f002:**
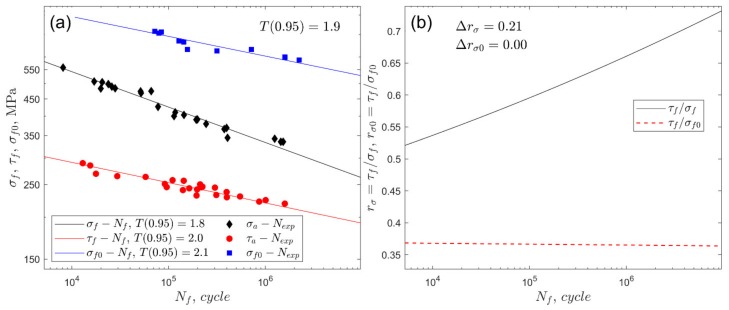
(**a**) Fatigue characteristics of the S355 steel for fully reversed bending $\sigma_{f}-N_{f}$, torsion $\tau_{f}-N_{f}$, and the zero-pulsating bending $\sigma_{f0}-N_{f}$ (*R* = 0). (**b**) Fatigue strength ratios $r_{\sigma }$ and $r_{\sigma 0}$.

**Figure 3 materials-13-01587-f003:**
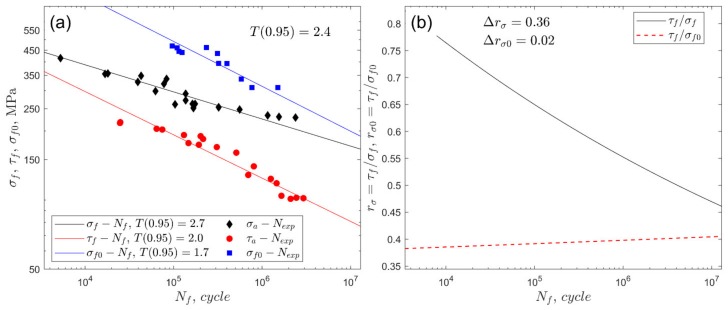
(**a**) Fatigue characteristics of the aluminium alloy 7075-T651 for fully reversed bending $\sigma_{f}-N_{f}$, torsion $\tau_{f}-N_{f}$, and the zero-pulsating bending $\sigma_{f0}-N_{f}$ (*R* = 0). (**b**) Fatigue strength ratios $r_{\sigma }$ and $r_{\sigma 0}$.

**Figure 4 materials-13-01587-f004:**
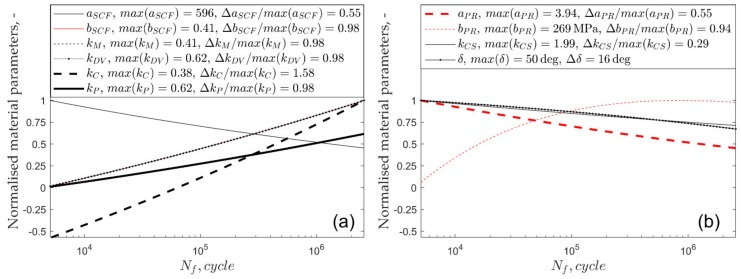
Normalised material parameters in the function of number of cycles to failure for S355 steel: (**a**) linear models and (**b**) non-linear models.

**Figure 5 materials-13-01587-f005:**
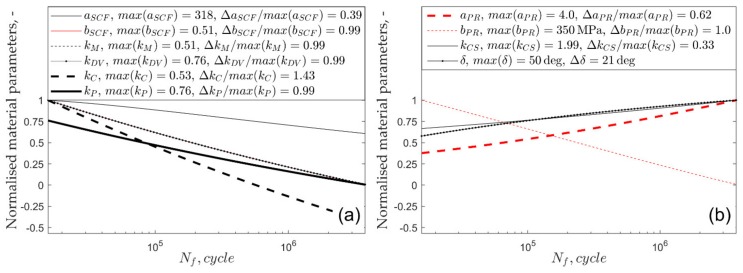
Normalised material parameters in the function of number of cycles to failure for aluminium alloy 7075-T651: (**a**) linear models and (**b**) non-linear models.

**Figure 6 materials-13-01587-f006:**
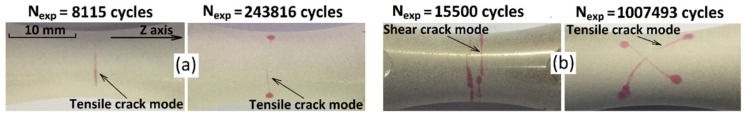
Exemplary photos of fatigue cracks under cyclic (**a**) bending and (**b**) torsion loads for the S355 steel.

**Figure 7 materials-13-01587-f007:**
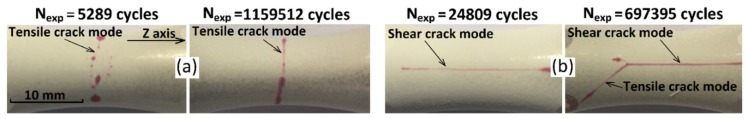
Exemplary photos of fatigue cracks under cyclic (**a**) bending and (**b**) torsion loads for the 7075-T651 aluminium alloy.

**Figure 8 materials-13-01587-f008:**
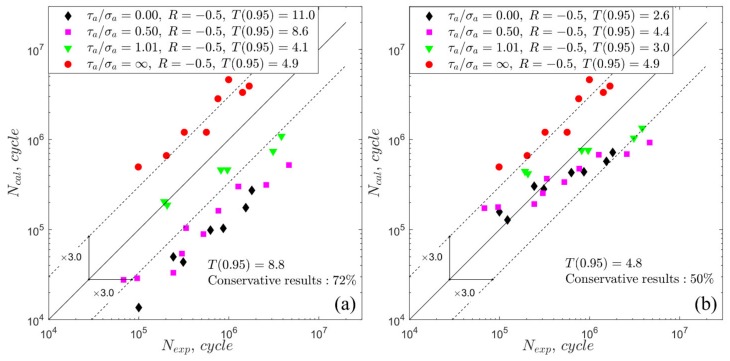
Experimental and calculated fatigue lives for the Crossland model under R = −0.5 for S355 steel: (**a**) fixed material parameters; (**b**) life-dependent material parameters.

**Figure 9 materials-13-01587-f009:**
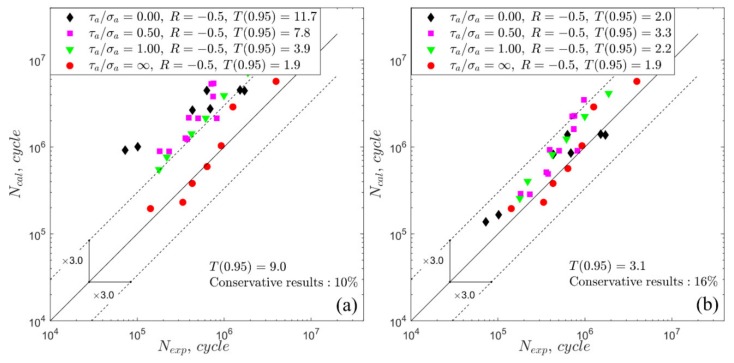
Experimental and calculated fatigue lives for the Crossland model under *R* = −0.5 for 7075-T651 aluminium alloy: (**a**) fixed material parameters; (**b**) life-dependent material parameters.

**Figure 10 materials-13-01587-f010:**
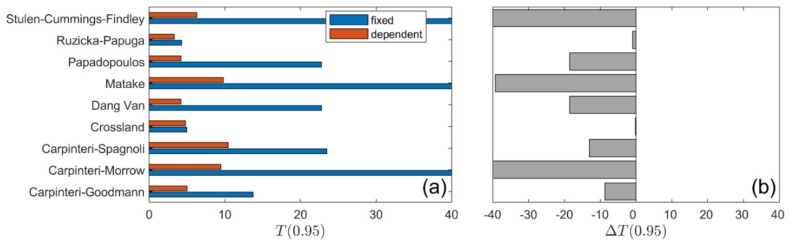
Summary comparison of calculation results for the analysed models under *R* = 0.0 for S355 steel: (**a**) fatigue scatter band and (**b**) change in the fatigue scatter band for all analysed fatigue models.

**Figure 11 materials-13-01587-f011:**
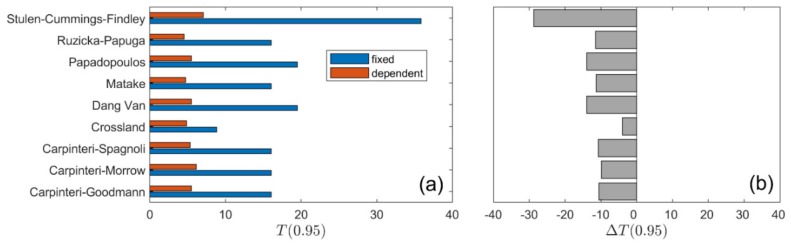
Summary comparison of calculation results for the analysed models under *R* = −0.5 for S355 steel: (**a**) fatigue scatter band and (**b**) change in the fatigue scatter band for all analysed fatigue models.

**Figure 12 materials-13-01587-f012:**
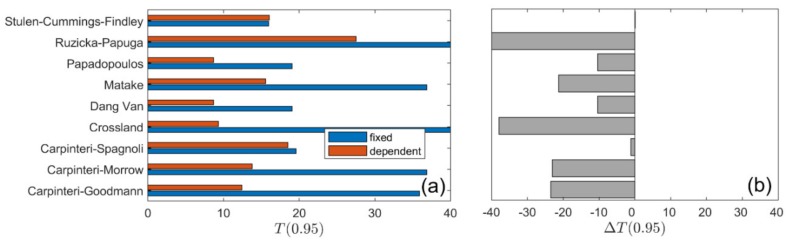
Summary comparison of calculation results for the analysed models under R = 0.0 for 7075-T651 aluminium alloy: (**a**) fatigue scatter band and (**b**) change in the fatigue scatter band for all analysed fatigue models.

**Figure 13 materials-13-01587-f013:**
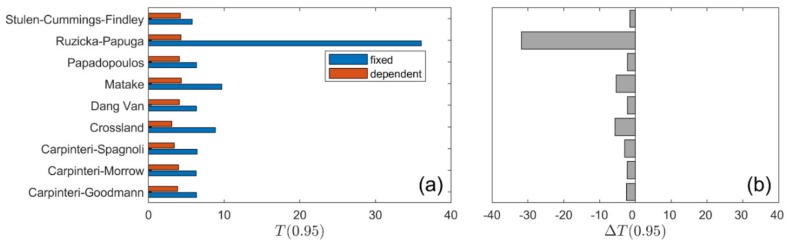
Summary comparison of calculation results for the analysed models under R = −0.5 for 7075-T651 aluminium alloy: (**a**) fatigue scatter band and (**b**) change in the fatigue scatter band for all the analysed fatigue models.

**Table 1 materials-13-01587-t001:** Selected models for fatigue life estimation with the life-dependent material parameters.

Model	Life-Dependent Material Parameters
Crossland [49] $\sqrt{J_{2,a}}+k_{C}\left(N_{f}\right)\sigma_{H,max}=\;\tau_{f}\left(N_{f}\right)$	$k_{C}=3r_{\sigma }-\sqrt{3}\;$
Stulen–Cummings–Findley [55,56] $\underset{n}{\max}\left\{a_{SCF}\left(N_{f}\right)\tau_{ns,a}+b_{SCF}\left(N_{f}\right)\sigma_{n,max}\right\}=\tau_{f}\left(N_{f}\right)$	$a_{SCF}=2\sqrt{r_{\sigma }+r_{\sigma }^{2}},\;\;b_{SCF}=2r_{\sigma }-1\;$
Dang Van [57] $\underset{n}{\max}\left\{\underset{t}{\max}\left\{\tau_{ns}\left(t\right)\right\}+k_{DV}\left(N_{f}\right)\sigma_{H,\max}\left(t\right)\right\}=\tau_{f}\left(N_{f}\right)$	$k_{DV}=3\left(r_{\sigma }-\frac{1}{2}\right)\;$
Matake [58] $\underset{n}{\max}\left\{\tau_{ns,a}\right\}+k_{M}\left(N_{f}\right)\sigma_{n,a}=\tau_{f}\left(N_{f}\right)$	$k_{M}=2r_{\sigma }-1$
Papadopoulos [50,51] $\underset{n}{\max}\left\{T_{a}\right\}+k_{P}\left(N_{f}\right)\sigma_{H,\max}=\tau_{f}\left(N_{f}\right)$	$k_{P}=3\left(r_{\sigma }-\frac{1}{2}\right)\;$
Carpinteri–Spagnoli [59] $\sqrt{\sigma_{n,max}^{2}+k_{CS}{\left(N_{f}\right)}^{2}\tau_{ns,a}^{2}}=\sigma_{f}\left(N_{f}\right)$	$k_{CS}=r_{\sigma }^{-1}$ $\delta =45\cdot \frac{3}{2}\left(1-r_{\sigma }{}^{2}\right)$
Papuga–Růžička [52] $\underset{n}{\max}\left\{\sqrt{a_{PR}\left(N_{f}\right)\tau_{ns,a}^{2}+b_{PR}\left(N_{f}\right)\left(\sigma_{n,a}+r_{\sigma 0}\left(N_{f}\right)\sigma_{n,m}\right)}\right\}=\sigma_{f}\left(N_{f}\right)$	$a_{PR}=\frac{r_{\sigma }^{-2}}{2}+\frac{\sqrt{r_{\sigma }^{-4}-r_{\sigma }^{-2}}}{2},\;b_{PR}=\sigma_{f}\;for\;r_{\sigma }^{-1}<\sqrt{\frac{4}{3}}$ $a_{PR}={\left(\frac{4r_{\sigma }^{-2}}{4+r_{\sigma }^{-2}}\right)}^{2},\;b_{PR}=\frac{8\sigma_{f}r_{\sigma }^{-2}\left(4-r_{\sigma }^{-2}\right)}{{\left(4+r_{\sigma }^{-2}\right)}^{2}}\;for\;r_{\sigma }^{-1}\geq \sqrt{\frac{4}{3}}\;$

**Table 2 materials-13-01587-t002:** Chemical composition of S355 steel (wt.%).

C	Mn	Si	P	S	Cr	Ni	Cu	Fe
0.21	1.46	0.42	0.019	0.046	0.09	0.04	0.17	Balance

**Table 3 materials-13-01587-t003:** Chemical composition of aluminium alloy 7075-T651 (wt %).

Mg	Mn	Fe	Si	Ti	Cu	Zn	Cr	Zr + Ti	Others Total	Others Each	Al
2.10–2.90	≤0.30	≤0.50	≤0.40	≤0.20	1.20–2.00	5.10–6.10	0.18–0.28	≤0.25	≤0.15	≤0.05	Balance

**Table 4 materials-13-01587-t004:** Basic mechanical properties for the analysed materials.

Material	$\sigma_{y}$ MPa	$\sigma_{u},$ MPa	E, MPa
S355	355	541	2.02 × 105
7075-T651	504	560	72 × 103

**Table 5 materials-13-01587-t005:** Regression coefficients of the fatigue characteristics with intervals for a 95% confidence level.

Material	Bending, *R* = −1		Torsion, *R* = −1		Bending, *R* = 0	
	*A* *_σ_* *_f_*	*m* *_σ_* *_f_*	*A* *_τ_* *_f_*	*m* *_τ_* *_f_*	*A* *_σ_* *_f0_*	*m* *_σ_* *_f0_*
S355	29.92 ± 2.39	9.48 ± 0.91	44.78 ± 6.23	16.55 ± 2.60	53.31 ± 14.32	17.01 ± 5.09
7075-T651	25.93 ± 4.06	8.56 ± 1.44	16.91 ± 1.69	5.20 ± 0.77	18.76 ± 4.35	5.11 ± 1.67

The axial fatigue strength coefficient is determined from $\sigma_{f}^{'}=10^{\left(A_{\sigma f}+\log2\right)/m_{\tau f}}$ (S355: $\sigma_{f}^{'}=1534\;\mathrm{MPa},$ 7075-T651: $\sigma_{f}^{'}=1160\;\mathrm{MPa}$).

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
