# Peer review of "Application of Life-Dependent Material Parameters to Fatigue Life Prediction under Multiaxial and Non-Zero Mean Loading"

_materials, 2020, doi:10.3390/ma13071587_

Round 1

Reviewer 1 Report

The aim of this work is verification of the life-dependent material parameters concept applied to multiaxial fatigue loading with non-zero mean stress. Morrow’s correction is employed to consider the effect of mean stress in the Carpinteri-Spagnoli model. However there are the following major issues:

  • What is the innovation in the paper? In particular, as also stated in line 123, the Goodman correction had already been applied to the Carpinteri-Spagnoli model. The necessity to employ Morrow’s correction is not discussed. Specially that according to Figures 12 and 13, previous models are working well.
  • How was the microstructural analysis linked to the fatigue life modeling?
  • The fact that mean shear stress has significant effect on the fatigue response is worth publishing. However, the paper’s scope is defined as the verification of life-dependent material parameters fatigue models.

For the above reasons, I do not recommend publication in the current format. The manuscript needs major revisions so that it clearly states whether the novelty is developing a new model (Morrow’s correction), or the paper is just employing and assessing previously developed models. Also, a link is required to be established between the microstructural analysis and the fatigue modeling.

Aside from these major issues on the novelty and contributions of the paper to learning something new, there are a few minor things as well:

  • A proof reading is highly recommended. There are some grammatical errors such as the one in line 142 (a extruded bar). Also, some figures’ and tables’ numbers should be changed: tables 1 & 2, Figures 2a and 3a in line 173; line 179: (b) is extra; line 141: Tables 2 and 3?
  • It is suggested that Table1 contains some other useful information as well; i.e. it summarizes lines 110 – 116. δ should be defined in the Table as well.
  • Some parts of the text is exactly repeated: lines 82 and 143
  • Are the life dependent material parameters in the model proposed in line 125 the same as model in Table 1?
  • Line 132: Specify clearly that the test is “pure shear”.
  • What is the frequency in the fatigue tests?
  • Figures 4b and 5b need clarifications: What are the red curves? Nothing has been mentioned in the text.
  • Describe how you have normalized the parameters in Figures 6 and 7
  • legends are difficult to be read in figures
  • In my opinion, microstructural analysis is nor relevant to the study. BTW, should exist a justification for it, it is recommended to label crack in Figure 8 and specify the cracking modes
  • Has dye penetration inspection carried out to reveal the surface cracks in Figure8? If yes, mention it.
  • You may discuss the critical planes as well to predict the cracking plane and compare it with the experimental observations in Figure 8
  • Explain how you determined 95% scatter bands, or refer to a paper/ ASTM standard.
  • Is the x-axis in figures 12 and 13 showing percentage?

Author Response

The comments given by the reviewers are very helpful to increase the quality of the manuscript. We are grateful for such professional remarks. All the comments are deeply analysed and appropriate corrections are introduced to the revised version of the manuscript. The changes made in the manuscript are marked.

Detailed responses to the Reviewer’ comments are given below.  

The aim of this work is verification of the life-dependent material parameters concept applied to multiaxial fatigue loading with non-zero mean stress. Morrow’s correction is employed to consider the effect of mean stress in the Carpinteri-Spagnoli model. However there are the following major issues:

  • What is the innovation in the paper? In particular, as also stated in line 123, the Goodman correction had already been applied to the Carpinteri-Spagnoli model. The necessity to employ Morrow’s correction is not discussed. Specially that according to Figures 12 and 13, previous models are working well.

Response 1:

The effectiveness of multiaxial fatigue models with the life-dependent material parameters under cyclic multiaxial with zero-mean stress was validated in a few research papers [23,37,39] (references according to the revised version of the manuscript) . For fatigue loading with the non-zero mean stress, the life-dependent material parameters concept was analysed only in [48]. The paper [48] was focused on the algorithm of lifetime calculation including with the concept of life-dependent material parameters. The proposed concept needs to be further verified on a larger group of materials and models. Additionally, the results of new experimental fatigue tests under uniaxial and multiaxial cyclic bending and torsion loading with stress ratio equal to R = -0.5 and R = 0.0 are presented. Several well-known stress-based fatigue life prediction models are implemented. For the first time, the life-dependent material parameter concept was introduced to the Crossland [49], Papadopoulos [50,51] and Papuga-Růžička [52] criteria under fatigue loading including non-zero mean stress.

The novelty of the research was highlighted on page 2 (Introduction), lines 70-83.

The Morrow correction was verified to check its effectiveness. Unfortunately, the effectiveness was not improved but it is important to present such result to eliminate this correction from further analysis.

  • How was the microstructural analysis linked to the fatigue life modeling?

Response 2:

The microstructural analysis is important at least for two reasons: (1) The material microstructure among other unapplied in presented research data such as the chemical composition and basic mechanical properties are important to characterised deeply the analysed materials. Thus, some other researchers could possible used this information for their research. (2) The microstructure analysis could explain some behaviour of a material under the fatigue loading. In the presented research, the observed anisotropy in the microstructure of 7075-T651 aluminium alloy could be reflected in the variability of material parameters. Additional explanations are presented in the revised manuscript on page 5, lines 163-173 and page 9, lines 241-244.

  • The fact that mean shear stress has significant effect on the fatigue response is worth publishing. However, the paper’s scope is defined as the verification of life-dependent material parameters fatigue models.

Response 3:

The significant effect of a non-zero mean of shear stress was highlighted in the manuscript (page 10, line 268-272, page 11, lines 297-298). This observation is the results of performed analysis and it was planned as the main aim of the research. The present paper aimed to verify the life-dependent material parameters concept under the multiaxial fatigue loading with the application of common stress-based models.

For the above reasons, I do not recommend publication in the current format. The manuscript needs major revisions so that it clearly states whether the novelty is developing a new model (Morrow’s correction), or the paper is just employing and assessing previously developed models. Also, a link is required to be established between the microstructural analysis and the fatigue modeling.

Response: 4

Answers are already given in the previous comments.

Aside from these major issues on the novelty and contributions of the paper to learning something new, there are a few minor things as well:

  • A proof reading is highly recommended. There are some grammatical errors such as the one in line 142 (a extruded bar). Also, some figures’ and tables’ numbers should be changed: tables 1 & 2, Figures 2a and 3a in line 173; line 179: (b) is extra; line 141: Tables 2 and 3?

Response: 5

The manuscript underwent extensive English editing performed by professional English MDPI editing service. The numbers of figures and tables are corrected.

  • It is suggested that Table1 contains some other useful information as well; i.e. it summarizes lines 110 – 116. δ should be defined in the Table as well.

Response: 6

After the submission of the manuscript, the editorial office has made some formatting changes to our original submission. This formatting introduced some unexpected modification of Table 1. Symbol δ has been already defined in the original submission.

  • Some parts of the text is exactly repeated: lines 82 and 143

Response: 7

The text was repeated due to the formatting discussed above. The revised manuscript is based on the doc file downloaded from the submission system and all bad formatting issues were corrected.

  • Are the life dependent material parameters in the model proposed in line 125 the same as model in Table 1?

Response: 8

Yes.

  • Line 132: Specify clearly that the test is “pure shear”.

Response: 9

The applied force is transferred to the specimen through the lever, so it is not pure shear but the additional shear stress is insignificant around 1% of the shear stress due to torque. The experimental fatigue tests are more deeply presented with information of applied loading. Page 4, lines 143-148.

  • What is the frequency in the fatigue tests?

Response: 10

Between 5-30 Hz depending on the applied stress amplitudes. The experimental fatigue tests are more deeply presented with information of applied loading. Page 4, lines 147-148.

  • Figures 4b and 5b need clarifications: What are the red curves? Nothing has been mentioned in the text.

Response: 11

The red lines present the variation of  related to the zero-pulsating bending loading. It was clarified on page 6, line 193.

  • Describe how you have normalized the parameters in Figures 6 and 7

Response: 12

The material parameters were normalised by their maximum value. It was clarified on page 7, line 206.

  • legends are difficult to be read in figures

Response: 13

The text included in legends and x and y labels were enlarged in figures 4-7 and 10-15.

  • In my opinion, microstructural analysis is nor relevant to the study. BTW, should exist a justification for it, it is recommended to label crack in Figure 8 and specify the cracking modes

Response: 14

The need for microstructural analysis was presented in response no 2. The figures 8 and 9 are modified to include labels, scale and Z-axis.

  • Has dye penetration inspection carried out to reveal the surface cracks in Figure8? If yes, mention it.

Response: 15

Yes. The appropriate description was added on page 8, lines 216-217.

  • You may discuss the critical planes as well to predict the cracking plane and compare it with the experimental observations in Figure 8

Response: 16

The problem of fracture plane orientations and its relation to the critical plane was not the subject of the present study.

  • Explain how you determined 95% scatter bands, or refer to a paper/ ASTM standard.

Response: 17

Equation (Eq. 7) with the definition of scatter band T and additional explanation were added on page 6, lines 186-188.

  • Is the x-axis in figures 12 and 13 showing percentage?

Response:

No.

Reviewer 2 Report

Review for Materials- 752453

Application of the life-dependent material parameters to fatigue life prediction for S355 steel and 7075-T651 aluminium alloy under multiaxial and non-zero mean loading

The authors address an interesting research topic for the journal Materials. In addition, the comparison of several mathematical fatigue models could generate lot of cites in the future. Furthermore, it is a rigorous and well-organized paper. Anyway, some recommendations should be considered:

  • Authors should be more careful with the MDPI format. Please revise the whole manuscript taking into account the MDPI format.
  • In my opinion, the title could be more general thereby attracting a greater number of readers, e.g. the following data could be deleted from the title “for S355 steel and 7075-T651 aluminium alloy”. This data could be included in the keyword.
  • Line 66: “Additional static loading could be an important factor affecting fatigue life”. A comment to the influence of fatigue on residual stresses is advisable. Authors could cite the related papers: https://doi.org/10.3390/app7010084 ; https://doi.org/10.3390/met9090966 ; https://doi.org/10.1016/j.engfracmech.2017.08.018
  • Please, improve the aesthetics of the Table 1 to facilitate the understanding of its content.
  • For a better comparison, use the same scale in vertical axis of Figures 6 and 7.
  • Line 234: “applies stress” => applied stress
  • For the sake of clarity, please use (a) and (b) in the figure caption of Figures 12, 13, 14 and 15.
  • Please, revise the format of the references according the MDPI guidelines (e.g. refs. 11, 12, 32, 37,…are incomplete)

Author Response

The comments given by the reviewers are very helpful to increase the quality of the manuscript. We are grateful for such professional remarks. All the comments are deeply analysed and appropriate corrections are introduced to the revised version of the manuscript. The changes made in the manuscript are marked.

Detailed responses to the Reviewer’ comments are given below.

The authors address an interesting research topic for the journal Materials. In addition, the comparison of several mathematical fatigue models could generate lot of cites in the future. Furthermore, it is a rigorous and well-organized paper. Anyway, some recommendations should be considered:

  • Authors should be more careful with the MDPI format. Please revise the whole manuscript taking into account the MDPI format.

Response 1:

The whole manuscript was revised and corrected according to required MDPI format.

  • In my opinion, the title could be more general thereby attracting a greater number of readers, e.g. the following data could be deleted from the title “for S355 steel and 7075-T651 aluminium alloy”. This data could be included in the keyword.

Response 2:

The title was modified according to the proposed corrections.

  • Line 66: “Additional static loading could be an important factor affecting fatigue life”. A comment to the influence of fatigue on residual stresses is advisable. Authors could cite the related papers: https://doi.org/10.3390/app7010084 ; https://doi.org/10.3390/met9090966 ; https://doi.org/10.1016/j.engfracmech.2017.08.018

Response 3:

Recommended articles were added on page 2, lines 69-70.

  • Please, improve the aesthetics of the Table 1 to facilitate the understanding of its content.

Response 4:

After the submission of the manuscript, the editorial office has made some formatting changes to our original submission. This formatting introduced some unexpected modification of Table 1. Table 1 was formatted again. We hope that this time it would not be modified.

  • For a better comparison, use the same scale in vertical axis of Figures 6 and 7.

Response 5:

Corrected. Additionally, fonts in legends and x, y labels were enlarged.

  • Line 234: “applies stress” => applied stress

The manuscript underwent extensive English editing performed by professional English MDPI editing service.

  • For the sake of clarity, please use (a) and (b) in the figure caption of Figures 12, 13, 14 and 15.

Response 6:

Corrected.

  • Please, revise the format of the references according the MDPI guidelines (e.g. refs. 11, 12, 32, 37,…are incomplete)

Response 7:

The whole manuscript was revised and corrected according to required MDPI format.

Reviewer 3 Report

First, the English language should be checked by a native English speaker. Sometimes the text is confusing and there are many "a" and "the" missing.

In line 83 a whole portion of the text is doubled. It should be deleted.

The authors discuss a lot the scatter bands and the T(0.95) parameter. This parameter is very important for most of their conclusions. This means that a determination of the scatter bands and the T(0.95) parameter should be explained in detail in the text. A reference to the literature is not enough.

There is a huge empty space between the lines 198 and 208.

Please, give the equations for parameters in figures 6 and 7 in the appendix.

Author Response

The comments given by the reviewers are very helpful to increase the quality of the manuscript. We are grateful for such professional remarks. All the comments are deeply analysed and appropriate corrections are introduced to the revised version of the manuscript. The changes made in the manuscript are marked.

Detailed responses to the Reviewer’ comments are given below.

First, the English language should be checked by a native English speaker. Sometimes the text is confusing and there are many "a" and "the" missing.

Response 1:

The manuscript underwent extensive English editing performed by professional English MDPI editing service.

In line 83 a whole portion of the text is doubled. It should be deleted.

Response 2:

After the submission of the manuscript, the editorial office has made some formatting changes to our original submission. This formatting introduced some unexpected modification of Table 1 and also part of the text was repeated. It was deleted.

The authors discuss a lot the scatter bands and the T(0.95) parameter. This parameter is very important for most of their conclusions. This means that a determination of the scatter bands and the T(0.95) parameter should be explained in detail in the text. A reference to the literature is not enough.

Response 3:

Equation (Eq. 7) with the definition of scatter band T and additional explanation were added on page 6, lines 186-188.

There is a huge empty space between the lines 198 and 208.

Response 4:

Deleted.

Please, give the equations for parameters in figures 6 and 7 in the appendix.

Response 5:

The equation for parameters presented in figures 6 and 7 are included in Table 1. Table 1 presented in the original submission also included those equations but due to some unexpected formatting by the editorial office, the presence of Table 1 became very unreadable.

Round 2

Reviewer 1 Report

The new manuscript has been improved. Some minor revisions are suggested:

  • You may discuss the determination T(0.95) in more details. This parameter has been used extensively in the paper and its calculation should be clearer to the reader.
  • While the supplementary word file is readable with no editorial error, the PDF file is corrupted in lines 132 and 140.
  • Label X, Y, and Z axes in all figures of 2 and 3; i.e., figures 2a,2b, and 3b
  • The link between the microstructure and fatigue analysis is still week. The early crack growth behavior can be used as a testimony for the necessity to employ life-dependent parameters. However, the microstructure analysis is not suggesting it. I agree that microstructure analysis is essential in characterizing the mechanics characteristic of materials. However, the discussion here is not supporting the subject of the paper. According to the manuscript, steel's microstructure is similar in different planes, yet aluminum’s is not. On the other hand, cracking inspection is suggesting different modes of cracking at different fatigue lives which is not supported by the microstructure. I still suggest to remove the microstructure section.
  • If you still want to keep the microstructure section: it is recommended to revise the section and discuss how microstructure suggests different cracking mode at different fatigue lives. Additionally, to better discuss microstructure-early crack growth-fatigue parameters relationships, present the microstructural analysis and early crack growth behavior sections back-to-back. In this format, a reader can better understand the necessity to employ the life-dependent parameters.

Author Response

The new manuscript has been improved. Some minor revisions are suggested:

You may discuss the determination T(0.95) in more details. This parameter has been used extensively in the paper and its calculation should be clearer to the reader.

Response 1:

The fatigue scatter band T(0.95) was calculated from empirical cumulative distribution of T. The shape-preserving piecewise cubic interpolation was applied to find T at 0.95 probability. Additional explanation is given on page 6, lines 186-191.

While the supplementary word file is readable with no editorial error, the PDF file is corrupted in lines 132 and 140.

Response 2:

We submitted only doc file. The pdf file was generated in the system and We did not have a possibility to verify the output.

Label X, Y, and Z axes in all figures of 2 and 3; i.e., figures 2a,2b, and 3b

Response 3:

Figures 2 and 3 were removed from the manuscript.

The link between the microstructure and fatigue analysis is still week. The early crack growth behavior can be used as a testimony for the necessity to employ life-dependent parameters. However, the microstructure analysis is not suggesting it. I agree that microstructure analysis is essential in characterizing the mechanics characteristic of materials. However, the discussion here is not supporting the subject of the paper. According to the manuscript, steel's microstructure is similar in different planes, yet aluminum’s is not. On the other hand, cracking inspection is suggesting different modes of cracking at different fatigue lives which is not supported by the microstructure. I still suggest to remove the microstructure section.

If you still want to keep the microstructure section: it is recommended to revise the section and discuss how microstructure suggests different cracking mode at different fatigue lives. Additionally, to better discuss microstructure-early crack growth-fatigue parameters relationships, present the microstructural analysis and early crack growth behavior sections back-to-back. In this format, a reader can better understand the necessity to employ the life-dependent parameters.

Response 4:

Dear Reviewer, thank you for the provided deep explanation and the your point of view. Although, the microstructure analysis is important specially from the fatigue point of view, it could mislead the readers from major intention of the research – implementation of the life-dependent material parameters to multiaxial fatigue life prediction models. The physical bases of this concept are still not fully recognized and they are under investigation. Thus, Figures 2, 3 and related text were removed from the revised manuscript.

Reviewer 2 Report

Review for Materials- 752453

Application of Life-Dependent Material Parameters to Fatigue Life Prediction under Multiaxial and Non-3 Zero Mean Loading

One minor comment:

  • Please, revise the lines 132 and 140: some equations with writing defects appear.

Author Response

One minor comment:

Please, revise the lines 132 and 140: some equations with writing defects appear.

Response 1:

The submitted manuscript did not include such defects. Anyway, it was corrected in the revised manuscript.

Reviewer 3 Report

1.) The PDF document is totally corrupted between equation (4) and (5). On the other hand, the DOC file is OK and readable.

2.) Explanation about calculating the T(0.95) parameter is still weak. The authors have provided the equation, in which they explained its calculation. However, which statistical distribution was applied to calculate the value of the T parameter for the 0.95 probability of what: survival or rupture? Was it the Student's T, normal, log-normal, Weibull's distribution? A reference to the past articles is not enough. I mean there exist methods for estimating the so called P-S-N curves, which estimate the parameters of the S-N curves and their scatter. See, for example, the articles for the Weibull's probability model:

  1. Klemenc, J. Influence of fatigue–life data modelling on the estimated reliability of a structure subjected to a constant-amplitude loading. Reliab Eng Syst Safe 2015, Volume 142, 238–247.

or ASTM standards for log-normal probability models.

Author Response

1.) The PDF document is totally corrupted between equation (4) and (5). On the other hand, the DOC file is OK and readable.

Response 1:

We submitted only doc file. The pdf file was generated in the system and We did not have a possibility to verify the output.

2.) Explanation about calculating the T(0.95) parameter is still weak. The authors have provided the equation, in which they explained its calculation. However, which statistical distribution was applied to calculate the value of the T parameter for the 0.95 probability of what: survival or rupture? Was it the Student's T, normal, log-normal, Weibull's distribution? A reference to the past articles is not enough. I mean there exist methods for estimating the so called P-S-N curves, which estimate the parameters of the S-N curves and their scatter. See, for example, the articles for the Weibull's probability model:

Klemenc, J. Influence of fatigue–life data modelling on the estimated reliability of a structure subjected to a constant-amplitude loading. Reliab Eng Syst Safe 2015, Volume 142, 238–247. or ASTM standards for log-normal probability models.

Response:

The fatigue scatter band T(0.95) was calculated from empirical cumulative distribution of T. The shape-preserving piecewise cubic interpolation was applied to find T at 0.95 probability. Additional explanation is given on page 6, lines 186-191.